# Explainable Error Detection in Integrated Circuits Image Segmentation via Graph Neural Networks

## Abstract

The nanoscale complexity of modern integrated circuits (ICs) and the low error tolerance in segmentation tasks pose significant challenges for automated quality control. While deep learning–based IC segmentation has advanced, most approaches still rely on manual inspection due to limited error interpretability. Existing CNN-based error detectors operate holistically on entire images, making it difficult to localize specific faults such as open or short circuits. We propose a novel, explainable error detection framework based on Graph Neural Networks (GNNs). By converting each connected component of a segmentation mask into a feature-annotated graph, our method enables localized reasoning and identification of segmentation errors through graph classification. This formulation allows the model to detect outlier components and precisely highlight erroneous regions, offering strong interpretability. Experiments across diverse IC layouts and imaging conditions demonstrate the robustness and generalizability of our approach, enabling accurate and interpretable error detection at the component level.

## 1 Introduction

The rapid advancement of semiconductor technology has led to increasingly complex integrated circuit (IC) designs with smaller feature sizes and higher integration density (Mack, 2011). Accurate segmentation of IC structures from scanning electron microscope (SEM) images is essential for applications such as failure analysis and hardware assurance (Huang & Jing, 2007; Cai et al., 2018; Wilson et al., 2022).

However, segmentation algorithms often produce errors that can significantly impact downstream function-level analysis, which relies on accurately segmented circuit structures (Zhang et al., 2016). The complex structures of modern ICs, including multi-level interconnects, varying material contrasts, and noise artifacts during SEM imaging, make accurate segmentation difficult (see Fig. 1 for examples). Moreover, segmentation errors, such as short and open circuits, can have subtle visual manifestations that are challenging to detect using conventional image analysis techniques (Doudkin et al., 2005; Lee & Yoo, 2008; Cheng et al., 2018; 2019a;b; Hong et al., 2019; Wilson et al., 2020; Yu et al., 2022). Therefore, the detection of such segmentation errors is a crucial task.

Correcting such errors typically requires manual visual inspection by experts. However, a single IC chip could have millions of SEM images, making manual review impractical, posing a major bottleneck for large-scale industrial deployment. In this paper, our main objective is to perform:

- *Error Detection* in (given) IC image segmentation.

In contrast to IC image segmentation, the problem of *error detection* is less studied. Zhang et al. (2023) has proposed a CNN-based automatic error detection. Such an approach is holistic in nature, i.e., full images are used for both training and testing. A decision is made on the feature representation of the whole image.

However, like the "spot the difference" game,[1] segmentation errors in IC images are typically local in nature (cf. Fig. 1), while holistic approaches offer limited explainability regarding which specific

---

[1] https://en.wikipedia.org/wiki/Spot_the_difference

circuit elements are erroneous. Furthermore, unlike natural images, the foreground components, such as metal lines (i.e., conductive metal pathways that interconnect transistors and other components on a chip), exhibit minimal variation in appearance features like color, intensity, texture, or shape. Instead, they are primarily defined by their structural topology and spatial arrangement. This observation motivates a graph-based approach to error detection, which naturally aligns with the structural characteristics of IC images.

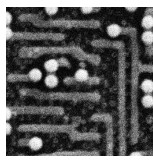 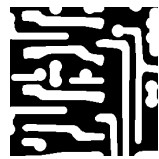 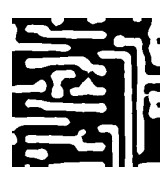

(a) Input IC image    (b) Ground-truth    (c) Predicted mask

Figure 1: For the IC image (a), its ground-truth segmentation mask is (b). The segmentation task is challenging due to noise and imaging artifacts, which may result in short or open circuits. Notably, the predicted mask (cf. (c)) often appears visually coherent, even when errors are present, making them difficult to detect through direct visual means. To address this, we propose a graph-based approach to identify local topological anomalies.

For the high-level idea, we encode each (metal-line) component $\mathcal{C}$ in an IC image or a binary mask using a graph $G$, which is a 1-dimensional skeleton of the component (see Fig. 2 and Fig. 4 below). The functionality of the component $\mathcal{C}$ is fully determined by its connectivity, and hence captured by the topology of $G$. We encode other information, such as position, thickness, and orientation, as features of $G$. Therefore, the error detection problem can be re-interpreted as a graph classification or outlier detection problem, for which we can employ Graph Neural Network (GNN) models (Defferrard et al., 2016; Kipf & Welling, 2017).

The proposed graph-based method offers several advantages. It provides inherent explainability by operating on individual connected components, allowing precise localization of segmentation errors. When an error is detected, the method can identify the specific nodes and edges in the graph that contribute to the classification decision. Moreover, it demonstrates superior generalization capability by focusing on topological features rather than image-level characteristics. This allows the method to handle datasets with varying image complexity and different numbers of connected components without requiring retraining. The graph-based representation is more robust (Hamilton et al., 2017) to variations in imaging conditions and noise artifacts that commonly affect SEM images.

Our contributions can be summarized as follows:

- Our work offers a novel graph-based perspective on IC image analysis, potentially paving the way for future research in this direction.

- We propose an efficient pipeline for converting a segmentation mask (for an IC) into a set of graphs. We describe how the mask information can be encoded with node and edge features.

- We design a tailored GNN model for our specific task. In particular, it leverages edge features for message passing in usual GNN models.

- We discuss why our approach is "explainable" and demonstrate that our approach can be applied in conjunction with computer vision (CV) models for the detection task.

## 2 PRELIMINARIES

### 2.1 PROBLEM: ERROR DETECTION IN IC IMAGE SEGMENTATION

Given an IC image $\mathcal{I}$, a segmentation model generates a binary mask $\mathbf{M}$ of the foreground components, most notably metal lines (see Fig. 4 left panel). For the *error detection problem*, we want to decide whether there is an error in the *given* binary mask $\mathbf{M}$, *without accessing* $\mathcal{I}$. Moreover, it is preferably able to pinpoint the error component/location with minimal human intervention.

As we have highlighted in Section 1, unlike natural images, the correctness of structural topology is a more important aspect for the functionality of an IC. Therefore, we propose to use graph-based methods, which are well-suited for such a need.

## 2.2 Graph neural networks

As outlined in Section 1, we shall encode foreground components as graphs with features, which can be further processed with graph neural networks (GNNs). In this subsection, we give a brief overview of GNNs.

Graph neural networks (GNNs) have emerged as powerful tools for analyzing structured data with complex relationships and topological properties (Defferrard et al., 2016; Veličković et al., 2017), which is particularly helpful for our setting. At the high level, many GNN models consist of layers of message-passing and feature aggregation (Xu et al., 2019). More specifically, given a graph $G = (V, E)$ and initial node features represented by $x_v^{(0)}$ for each $v \in V$, the node features can then be updated in the $l$-th layer as follows:

$$x_v^{(l)} = \sigma\big(W^{(l)}\text{AGGR}(\{x_u^{(l-1)} \mid u \in \mathcal{N}(v)\})\big) \tag{1}$$

where $\sigma$ is an activation function, $W^{(l)}$ are the learnable weights in the $l$-th layer and $\mathcal{N}(v)$ is the neighbor set of $v$. AGGR is a message aggregation function, such as a weighted average (Kipf & Welling, 2017) or a weighted sum (Xu et al., 2019). The latter is the backbone of the graph isomorphism network (GIN). For a *graph-level task*, one may retrieve a graph feature by applying a global pooling function (e.g., summation, average) to the output node features of the last layer.

As a preview, in the context of IC images, graph edges approximate metal lines and possess inherent attributes. To accommodate this, we refine the general formulation in (1) by integrating edge features, thereby constructing a model specifically tailored to the IC image analysis task.

# 3 Methodology

## 3.1 Why the graph approach: an overview

In this subsection, we outline the motivation for adopting a graph-based approach and provide a high-level overview of our model. To maintain clarity and avoid delving into technical concepts from metric geometry (Bridson & Haefliger, 1999) and algebraic topology (Hatcher, 2001), we keep the discussion informal here. A detailed and rigorous theoretical treatment is provided in Appendix A.

For a mask $\mathbf{M}$ of an IC image, a (metal-line) component $\mathcal{C}$ is usually regular in shape. Therefore, many of its essential functionalities, such as connectivity, can be captured with an embedded 1-dimensional metrical graph (i.e., a graph with a metric) $G \subset \mathcal{C}$. To be more specific, for a small $\epsilon > 0$, we call $G$ an $\epsilon$-approximation of $\mathcal{C}$ if (a) points on $\mathcal{C}$ are within $\epsilon$ distance to points in $\mathcal{G}$; (b) $\mathcal{C}$ can be continuously deformed to $\mathcal{G}$ without breaking (see Fig. 2).

Therefore, if we approximate each component in ground-truth mask $\mathbf{M}$ by a metrical graph, we obtain an arrangement of a collection of graphs $\mathcal{G}_\mathbf{M} = \{G_i \mid 1 \leq i \leq c\}$, where $c$ is the number of components of $\mathbf{M}$. Intuitively, consider a segmentation mask $\mathbf{M}'$ and let $\mathcal{G}_{\mathbf{M}'} = \{G_i' \mid 1 \leq i \leq c'\}$ be the collection of graphs similarly derived from $\mathbf{M}$. If $\mathbf{M}'$ contains errors such as open or short circuits, then some graph $G_i' \in \mathcal{G}_{\mathbf{M}'}$ does not have a comparable counterpart in $\mathcal{G}_\mathbf{M}$ (see Fig. 2), i.e., being *outliers* in reference to $\mathcal{G}_\mathbf{M}$.

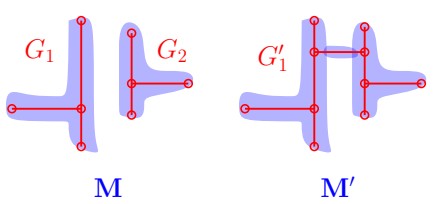

$\mathbf{M}$        $\mathbf{M}'$

Figure 2: In these examples, we have graphs the components of the masks. If a short circuit occurs for the segmentation mask $\mathbf{M}'$, then the graph $G_1'$ is not comparable to any graph derived from $\mathbf{M}$.

In other words, suppose $\mathbf{M}'$ is the segmentation mask and $\mathbf{M}$ is the ground-truth mask $\mathbf{M}$. A graph $G_i'$ in the graph collection $\mathcal{G}_{\mathbf{M}'}$ is *comparable* to $G_j$ in the ground-truth graph collection $\mathcal{G}_\mathbf{M}$ if the following holds:

(a) $G_i'$ and $G_j$ are located close to each other.

(b) $G_i'$ and $G_j$ are similar as both topological spaces and metric spaces.

If we do *not* have a one-to-one correspondence between $\mathcal{G}_{\mathbf{M}}$ and $\mathcal{G}_{\mathbf{M}'}$ of comparable graph pairs, then $\mathbf{M}'$ is likely to have errors such as open or short circuits. A more rigorous formulation and its proof are provided in Appendix A.

The upshot is that if no errors (e.g., open or short circuits) are present, then the graphs in the mask $\mathbf{M}'$ are likely to have been "seen before". If, across a dataset, graphs derived from ground-truth masks of different images follow a similar distribution, error detection can then be framed as identifying "unseen graphs", effectively reducing the task to a two-class classification problem.

To encode the "comparability" conditions defined above, the following features of component graphs are potentially useful for classification:

- locations (for positional proximity),

- lengths of edges and turning angles (for similarity in topology and geometric shape).

Additional visual information can be helpful, and more details are given in the next subsection.

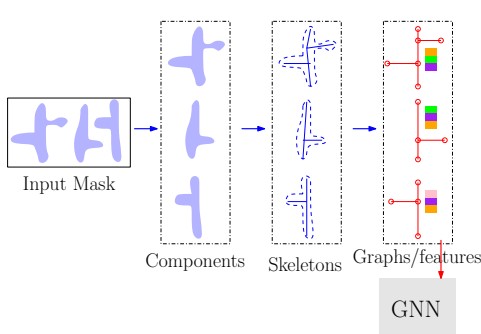

Input Mask    Components    Skeletons    Graphs/features

GNN

Figure 3: An input mask is converted into graphs with features, which are fed into a GNN. The figure summarizes the pipeline. An explicit example is given in Fig. 4.

The above discussions inspire us to adopt a graph-based strategy for error detection by framing the task as a classification problem over feature-annotated graphs, each derived from a connected component of the binary mask. It offers *explainability* by directly indicating anomalous components. This approach naturally decomposes into two core modules: (a) image-to-graph conversion, and (b) graph classification via GNNs, which we detail in the following subsections. A visual summary of our model pipeline is in Fig. 3.

### 3.2 IMAGE-TO-GRAPH CONVERSION

Suppose $\mathcal{I}$ is an IC image of dimension $H \times W$. Given a binary segmentation mask $\mathbf{M} \in \{0, 1\}^{H \times W}$, we extract individual connected components using 8-connectivity analysis. This means that two pixels $p_1, p_2$ are *adjacent* if $p_2$ is one of the 8 neighbors (including 4 diagonal neighbors) of $p_1$. They belong to the same component if there is a sequential path of pixels from $p_1$ to $p_2$ consisting of adjacent pixels.

Each component $\mathcal{C}$ represents a distinct metal-line structure. We apply the morphological thinning operation with the skeletonization algorithm in Zhang & Suen (1984), to reduce each component $\mathcal{C}$ to single-pixel-width medial axis representations $\mathcal{S}$ while preserving topological connectivity (see Fig. 4).

However, each skeleton $\mathcal{S}$ usually consists of a large collection of pixels, which does not represent a proper graph structure. To better represent the rectilinear nature of integrated circuit layouts, we convert the skeletons $\mathcal{S}$ into a graph $G = (V, E)$ with a small node set $V$ as follows.

**The node set** $V$    We categorize $V$ into three sub-types:

- *End points*. These are pixels with *exactly one neighbor* belonging to the skeleton $\mathcal{S}$ (among 8-neighbors). Intuitively, they represent the end points of the graph $G$.

- *Junctions*. These are pixels with *three or more neighbors* belonging to $\mathcal{S}$. Intuitively, they are connection points for different branches.

- *Corners*. These are pixels with *two neighbors* where the path direction makes a significant turn (see details below). Intuitively, they represent the turning corner locations of the metal line (see Fig. 4 right panel).

Figure 4: In the example, we are given the binary segmentation mask $\mathbf{M}$ (left panel). Skeletonization generates the skeletons of the components (middle panel), each of which contains an excessive number of nodes. The refinement greatly reduces the number of nodes and yields the final graph structures (red nodes, blue edges in the right panel).

*More on corners.* For a pixel $p_0$ with two neighbors, let $\mathcal{N}$ be the $g \times g$ neighborhood grid of $p_0$. The skeleton $\mathcal{S}$ intersects the boundary of $\mathcal{N}$ at two pixels $p_1, p_2$. We compute the angle $\theta$ between the vectors $\overrightarrow{p_0 p_1}$ and $\overrightarrow{p_0 p_2}$. The pixel $p_0$ is a *corner* if $\theta$ is within $90° \pm \Delta°$. Large $\theta$ indicates that there is no directional change, while a very small $\theta$ is unlikely, as meta-lines rarely reverse direction abruptly. Our implementation chooses $g = 5$ and $\Delta = 30$.

**The edge set** $E$ Edges are constructed by tracing paths between nodes in $V$ using *breadth-first search* on the skeleton $\mathcal{S}$. This prevents the connection of far-away nodes and thus faithfully captures the topology of the component.

The edge construction skips many topologically unimportant pixels. To compensate, we encode their information as features, which we discuss next.

**Node and edge features** For each node $v \in V$, we concatenate the following numerical information as a *node feature* vector $\mathbf{h}_v^{(0)} = [x_v, y_v, z_v, \boldsymbol{\theta}_v]$:

- $(x_v, y_v)$ is the coordinate of the pixel location, normalized with the image dimension.
- $z_v$ is the one-hot encoding of node type: end point, junction, or corner.
- The angles between edges incident at $v$ are recorded in a clockwise direction as a vector $\boldsymbol{\theta}_v$ of dimension $d$. Here, $d$ is a prescribed upper bound on the number of incident edges (usually $d = 4$ is sufficient). We use $0$ padding if the number of incident edges at $v$ is less than $d$.

As we have pointed out, it is crucial to use *edge features* to compensate for the information loss in graph conversion. For an edge $(v_i, v_j) \in E$, its feature vector

$$\mathbf{e}_{ij} = [\ell_{ij}, \mu_{ij}, \nu_{ij}, M_{ij}, m_{ij}] \tag{2}$$

encodes geometric and width information as follows:

- $\ell_{ij}$ is the Euclidean distance between the pixels for $v_i, v_j$.
- For a number $k$, we identify equally spaced $k$-points $P = \{p_1, \ldots, p_k\}$ on the edge $(v_i, v_j)$. The width of the binary mask perpendicular to $(v_i, v_j)$ and passing through each $p \in P$ is recorded as a vector $\mathbf{w}_{ij} = (w_1, \ldots, w_k)$ (see Fig. 5 for an example). Then $\mu_{ij}, \nu_{ij}, M_{ij}, m_{ij}$ are the average, variance, max and min of $\mathbf{w}_{ij}$, respectively.
  Intuitively, these features encode visual information of the original binary mask.

**Remark 1.** *The converted graphs may have small artifacts, e.g., small addendum edges, inherited from skeletonization. However, as such an edge has a short length, its contribution during feature aggregation almost vanished due to the continuous nature of the GNN. Therefore, such an artifact does not pose a serious challenge.*

In the next subsection, we describe how to incorporate node and edge features into a GNN for the error detection task.

### 3.3 EDGE-AWARE GRAPH ISOMORPHISM NETWORK

As in the "overview" subsection, we cast the explainable error detection problem as a classification problem of feature-annotated graphs with two types, either normal or abnormal. This is slightly different from the 1-class classification for anomaly detection (Zhao & Akoglu, 2021; Qiu et al., 2022), as we have both positive and negative training samples.

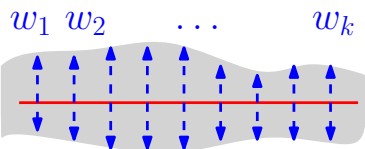

$w_1 \, w_2 \qquad \cdots \qquad w_k$

Figure 5: As shown in the example, the perpendicular width $w_1, \ldots, w_k$ of different locations on the skeleton (red) are used to form edge features.

The GIN model is proven to be a powerful model for graph classification with node features. Our tailor-made model is in the style of GIN. To highlight the incorporation of edge features, we call our model *Edge-aware Graph Isomorphism Network (EA-GIN)*.

An essential design requirement is to incorporate the edge features $\mathbf{e}_{ij}$ (2) into the model. A natural idea is to convert them into attention weights for node feature aggregation in message passing. Therefore, the $l$-th EA-GIN layer performs the following updates for the node $v_i$:

$$a_{ij}^{(l)} = \text{MLP}_1^{(l)}(\mathbf{e}_{ij}), \tag{3}$$

$$\widehat{a}_{ij}^{(l)} = \mathbf{D}_{ii}^{-1/2} \cdot a_{ij}^{(l)} \cdot \mathbf{D}_{jj}^{-1/2}, \tag{4}$$

$$\mathbf{h}_i^{(l+1)} = \text{MLP}_2^{(l+1)}\left( (1 + \epsilon^{(l)})\mathbf{h}_i^{(l)} + \sum_{j \in \mathcal{N}(i)} \widehat{a}_{ij}^{(l)} \mathbf{h}_j^{(l)} \right). \tag{5}$$

In the model, we use two learnable multilayer perceptrons (with ReLU activation): $\text{MLP}_1^{(l)}$ converts edge features into attention weights $a_{ij}^{(l)}$, and $\text{MLP}_2^{(l+1)}$ updates node representations. The parameter $\epsilon^{(l)}$ is learnable as in GIN. The weights $a_{ij}^{(l)}$ are normalized by the diagonal entries of the degree matrix $\mathbf{D}$ of the graph. As usual, the aggregation is performed in the 1-hop neighbor $\mathcal{N}(i)$ of $v_i$.

For a graph-level downstream task, a READOUT function (e.g., summation or another pooling function (Ying et al., 2018)) aggregates node features of the last layer $\{\mathbf{h}_i^{(L)} \mid v_i \in V\}$ to generate the graph representation:

$$\mathbf{h}_G = \text{READOUT}(\{\mathbf{h}_i^{(L)} \mid v_i \in V\}). \tag{6}$$

In our case, $\mathbf{h}_G$ is used for a two-class classification. It is passed through an MLP to generate two probability weights for the classes. The predictions are further used in the standard *cross-entropy loss* or *weighted cross-entropy loss* (when there is label imbalance) for training.

Although the model is not equipped with a complex mechanism, it seamlessly integrates all key elements. Its effectiveness is demonstrated through numerical results in the next section.

## 4 EXPERIMENTAL RESULTS

### 4.1 DATASETS AND EXPERIMENT OVERVIEW

Our model is evaluated across four IC image datasets: A1, A2, S1, and S2. Each dataset contains binary segmentation masks, ground-truth masks, and ground-truth labels for error detection. The datasets exhibit varying design characteristics in terms of circuit topology, feature density, and error types. In Table 1, we show the size of positive (i.e., with errors) and negative samples in each dataset. We notice that all datasets are highly class-imbalanced, and we evaluate model performance with the *F1-score*.

We perform the following studies in the subsections below:

- We combine graph conversion and EA-GIN as a full-fledged error detection model in IC image segmentation. We evaluate its performance on error detection by comparing it with several similarly constructed GNN-based detection models.
- We compare the performance of EA-GIN with CV (Zhang et al., 2023) and dedicated anomaly detection models (Qiu et al., 2022), and study their ability for cross-data generalization (i.e., training and testing performed on different datasets).
- We study the explainability of EA-GIN by analyzing its ability to pinpoint error locations, and explore the possibility of using EA-GIN to assist a CV-based model.

Table 1: Sizes of positive (i.e., with errors) and negative samples in each IC image dataset (img.) and the converted graph (graph) dataset.

|    | +ve img. | -ve img. | +ve graph | -ve graph |
|----|----------|----------|-----------|-----------|
| A1 | 190      | 31197    | 921       | 170609    |
| A2 | 102      | 29277    | 721       | 116602    |
| S1 | 10526    | 22643    | 22310     | 376329    |
| S2 | 2575     | 37269    | 6822      | 144335    |

## 4.2 ERROR DETECTION IN IC IMAGE SEGMENTATION: GNN MODELS

**Details for EA-GIN**    For our model, we use 3 EA-GIN layers (3)-(5) for feature aggregation. In the edge encoder (3), $MLP_1$ has one hidden layer with 8 hidden units (and ReLU activation). On the other hand, $MLP_2$ in (5) has one hidden layer with 64 hidden units. The "Summation" function is used as the READOUT function in (6). Finally, due to label imbalance, we use the weighted cross-entropy for training (Ling & Sheng, 2011) by assigning a higher weight to positive samples.

**Experimental setting and evaluation protocol**    A dataset consists of multiple images with a segmented binary mask, and we retrieve multiple graphs from each mask. As described earlier, each graph is converted from a connected component in the segmentation mask. We obtain the ground-truth label for the graphs from the ground-truth segmentation mask.

For the downstream error detection task in IC image segmentation, all (component) graphs for a single image should be collectively used either for training, validation or testing. Therefore, for each dataset (i.e., A1, A2, S1, S2), we split all images into training/validation/testing sets such that the corresponding split of graphs is *approximately* 70%/20%/10%.

We train EA-GIN and tune hyperparameters with the training/validation set. For testing, given a test image $\mathcal{I}$ with segmentation mask $\mathbf{M}$, we apply EA-GIN to the set $\mathcal{G}_{\mathbf{M}}$ of component graphs of $\mathbf{M}$. The segmentation is deemed erroneous if there is at least one graph in $\mathcal{G}_{\mathbf{M}}$ of error type. The resulting model is also called EA-GIN. The F1-score on the testing set is used as the evaluation metric.

The experiments are performed on a server with GPU: NVIDIA RTX A5000, 24GB memory.

**Results**    We compare EA-GIN with our backbone model GIN (Xu et al., 2019), and GNN models using *edge information*: GINE (Brossard et al., 2020), EGAT (Wang et al., 2021), and CensNet (Jiang et al., 2019). For any benchmark GNN model, the same procedure described above is applied for the downstream error detection task.

The results are shown in Table 2. We see that the models generally perform better on the A1 and A2 datasets. The IC structures in these two datasets are simpler with less variety. It is observed that our EA-GIN performs much better than its backbone GIN. Moreover, GINE also has a relatively good performance. Therefore, it is beneficial to incorporate edge features into the models.

EA-GIN generally outperforms benchmarks by a clear margin, and in the following subsection, we study other aspects of the model as outlined in Section 4.1.

Table 2: Results (F1-score) on error detection in IC image segmentation. The best performance is **boldfaced** and the 2nd best is underlined.

|         | A1         | A2         | S1         | S2         |
|---------|------------|------------|------------|------------|
| GIN     | 0.6274     | 0.8089     | 0.7316     | 0.6853     |
| GINE    | 0.9356 | 0.9498 | 0.7825 | 0.7614 |
| EGAT    | 0.4316     | 0.7503     | 0.4591     | 0.4401     |
| CensNet | 0.6458     | 0.8551     | 0.7189     | 0.6243     |
| EA-GIN  | **0.9884** | **0.9953** | **0.9183** | **0.8990** |

Table 3: Comparison of error detection models for IC image segmentation. The performance is measured by the F1-score. Cross-data generalization results are also reported.

| Train. | Test. | ED-ResNet | OCGTL | EA-GIN |
|---|---|---|---|---|
| A1 | A1 | 0.9823 | 0.9877 | **0.9884** |
|    | A2 | 0.9956 | 0.9961 | **0.9962** |
|    | S1 | 0.6598 | 0.8699 | **0.9070** |
|    | S2 | 0.9118 | 0.8762 | **0.9369** |
| A2 | A1 | 0.9534 | 0.8979 | **0.9827** |
|    | A2 | 0.9872 | 0.9270 | **0.9953** |
|    | S1 | 0.6543 | 0.8505 | **0.9396** |
|    | S2 | 0.9261 | 0.8650 | **0.9385** |
| S1 | A1 | **0.9726** | 0.9093 | 0.9580 |
|    | A2 | 0.9936 | **0.9950** | 0.9775 |
|    | S1 | **0.9795** | 0.8615 | 0.9183 |
|    | S2 | **0.9732** | 0.8805 | 0.9155 |
| S2 | A1 | **0.9666** | 0.7640 | 0.8951 |
|    | A2 | **0.9939** | 0.9273 | 0.9887 |
|    | S1 | 0.7657 | **0.8516** | 0.8230 |
|    | S2 | **0.9951** | 0.8842 | 0.8990 |

## 4.3 CROSS-DATA GENERALIZATION: CV AND ANOMALY DETECTION MODELS

Cross-data generalization plays a key role in hardware assurance, as ground-truth is often obtained from different batches of data in practice. This underscores the need for *robust* models. For the study, we compare with the CV benchmark ED-ResNet[2] dedicated to the error detection problem (Zhang et al., 2023). As the problem is closely related to anomaly detection, we also compare it with the graph anomaly detection model OCGTL (Qiu et al., 2022).[3] We show test results and cross-data generalization results in Table 3.

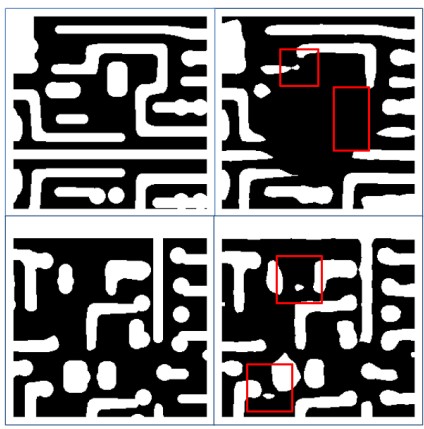

Figure 6: Examples of segmentation missed by ED-ResNet, while identified by EA-GIN. Left panels: ground-truth, right panels: segmentation mask, and error locations detected by EA-GIN are boxed.

From the results, we see that the CV-approach ED-ResNet and our graph approach EA-GIN have their respective advantages. ED-ResNet performs well when trained on larger and more complex datasets, whereas EA-GIN demonstrates strong generalization even when trained on smaller datasets, for example, when trained on A1 or A2 and evaluated on the larger dataset S1. This may be because the number of metal-line connection structures/patterns is limited, and hence, a small dataset may have given enough component graphs to capture feasible metal-line topologies. On the other hand, a CV model that requires a large amount of visual information may need more training data to identify essential features for detection.

For EA-GIN, it is interesting to notice that to test on S2, training on A1 or A2 yields a better result than training on S2 itself. This suggests that sufficiently many positive and negative patterns may be seen in A1 or A2, while training on the much larger S2 may overfit the model to a certain extent.

---

[2]We call it error detection with ResNet, abbreviated ED-ResNet for convenience.

[3]EA-GIN and ED-ResNet have a slight edge over OCGTL, as anomaly detection is one-class classification and does not utilize the small set of positive samples.

For illustration, we show examples of segmentation missed by ED-ResNet, while identified by EA-GIN in Fig. 6. It is observed that EA-GIN can capture small errors resembling noise, and this may be due to the local nature of the graph approach. The CV and graph approaches potentially contribute to the error detection problem in different ways, supplementing each other. Therefore, it can be beneficial to combine them, which we shall study in the next subsection.

### 4.4 EXPLAINABILITY

An important feature of the graph-based approach is its inherent explainability. This is particularly useful in practice to assist manual checking. In this study, we consider the following sequential merging of the CV and graph-based models. Specifically, for a dataset, we consider the set $\mathcal{S}^+$ of all test images correctly predicted to be erroneous by the CV model ED-ResNet. For each image in $\mathcal{S}^+$, we use EA-GIN to generate the probability score of a component being abnormal. The components with the top-$\kappa$ highest scores are flagged for further manual checking.

Table 4: F1-scores for the detection of error components with varying $\kappa$.

| $\kappa$ | A1 | A2 | S1 | S2 |
|---|---|---|---|---|
| 1 | 1.00 | 1.00 | 0.9249 | 0.9187 |
| 2 | 1.00 | 1.00 | 0.9492 | 0.9774 |
| 3 | 1.00 | 1.00 | 0.9724 | 0.9911 |

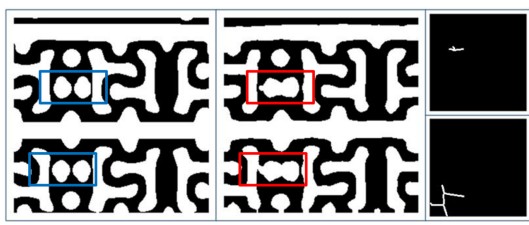

Figure 7: We show error localization of the examples in Fig. 1. The (highlighted) error components in the segmentation mask can be identified via their converted graphs. Left panels: ground-truth masks, middle panels: segmentation masks, right panels: skeletons of detected error components.

In Table 4, we show the results, measured by F1-scores, for whether the above procedure can correctly identify the error components when the budget $\kappa = 1, 2$ or $3$. We see that for A1 and A2, the graph approach can always correctly identify the error components, conditioned on knowing that the image contains errors. While for S1 and S2, increasing $\kappa$ to 3 yields reasonably good detection performance.

We show examples of the error localization in Fig. 7, in which the error components are correctly identified. In conclusion, our graph-based approach can be used as a stand-alone model or together with CV tools for automated error detection.

### 5 CONCLUSIONS

We propose an explainable graph-based framework for error detection in IC images segmentation, addressing key limitations of prior methods. Our approach introduces (i) a mask-to-graph conversion pipeline that encodes topological and geometric information into feature-annotated graphs, and (ii) a tailored EA-GIN model that incorporates edge features into the powerful GIN model for accurate graph classification. The method achieves strong cross-dataset performance and precise component-level error localization, offering scalability, robustness, and interpretability.

Future work includes enhancing graph representations with hierarchical or semantic information, exploring graph transformers, and integrating spatial cues from raw images. Real-time inference, uncertainty quantification, and adaptation to domains such as biological or road networks present promising directions, as does the development of interactive tools for quality assurance in manufacturing.

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

## A  THEORETICAL FRAMEWORK

In this appendix, we establish a theoretical framework for the study of IC segmentation masks. We employ formal mathematical concepts, enabling us to study IC segmentation-related problems rigorously and systematically.

### A.1  GEOMETRY AND TOPOLOGY FOR IC SEGMENTATION MASKS

For an IC image $\mathcal{I}$ of size $H \times W$, we model a *segmentation mask* $\mathbf{M}$ as a binary function (on the continuous domain):

$$\mathbf{M} : [0, H] \times [0, W] \to \{0, 1\}$$

such that the following holds: $\mathbf{M}^{-1}(1)$ has *finitely many* compact connected components, each with continuous boundaries.

We use $\mathcal{C}_{\mathbf{M}} = \{C_1, \ldots, C_m\}$ to denote the set of components of $\mathbf{M}^{-1}(1)$. They correspond to the set of metal-line components of the image.

To proceed, we need a few important concepts from metrical geometry Bridson & Haefliger (1999).

**Definition 1.** *In a metric space* $(X, d)$*, the* Hausdorff distance *between two subsets* $S_1, S_2$ *is*

$$d_H(S_1, S_2) = \max\{ \sup_{x_1 \in S_1} d(x_1, S_2), \sup_{x_2 \in S_2} d(S_1, x_2)\}.$$

*For two metric spaces* $(X_1, d_1)$ *and* $(X_2, d_2)$*, their* Gromov-Hausdorff *distance is defined as:*

$$d_{GH}(X_1, X_2) = \inf_{f, g, M} d_H(f(X_1), g(X_2)),$$

*where the infimum is taken over isometric embeddings of* $X_1, X_2$ *into the same metric space* $M$ *as:* $f : X_1 \to M$ *and* $g : X_2 \to M$.

For $\epsilon > 0$, we call $\mathbf{M}$ or $\mathcal{C}_{\mathbf{M}}$ $\epsilon$-*separated* if for $i \neq j \leq m$, $d_H(C_i, C_j) > \epsilon$. Intuitively, the conditions imply that the metal-line components on the segmentation mask are well-separated without ambiguity.

For each component $C_i$, an embedded subgraph $G_i \subset C_i$ (i.e., a subset homeomorphic to a graph) is an $\epsilon$-*approximation* for some (small) $\epsilon > 0$ if the following holds:

- Consider $G_i, C_i$ as subsets of $[0, W] \times [0, H]$ with the Euclidean metric:

$$d_H(G_i, C_i) \leq \epsilon;$$

- $G_i$ is a *homotopy retract* of $C_i$ (Hatcher, 2001).

Intuitively, the two conditions ensure the metrical and topological fidelity of $G_i$.

Fix $\epsilon > 0$. For each $C_i$, let the graph $G_i$ be an $\epsilon$-approximation. We use $\mathcal{G}_{\mathbf{M}}$ to denote the set $\{G_1, \ldots, G_m\}$.

We next formalize *open and short circuits*. For this, we assume $\mathbf{M}$ is the ground-truth mask and $\mathbf{M}'$ is a segmentation mask (to be tested). Accordingly, we have $\mathcal{C}_{\mathbf{M}'} = \{C'_1, \ldots, C'_{m'}\}$ and $\mathcal{G}_{\mathbf{M}'} = \{G'_1, \ldots, G'_{m'}\}$, which are defined similarly.

Fix a small threshold $\epsilon > 0$. We say that $\mathbf{M}'$ *does not* have *open or short circuits* if the following holds: for any $C_i \in \mathcal{C}_M$ and $C'_j \in \mathcal{C}_{M'}$ such that $C_i \cap C'_j \neq \emptyset$, then each connected component of their symmetric difference $C_i \Delta C'_j = (C_i \backslash C'_j) \cup (C'_j \backslash C_i)$ has diameter bounded by $\epsilon$.

Notice that for different concepts, the "error parameter $\epsilon$" is used independently in the respective definition. However, they may be related to each other if we bring all the concepts together.

Regarding open and short circuits, intuitively, as the masks are $\epsilon$-separated, a large component in the symmetric difference can only occur if there are open or short circuits.

Aside from the above common errors affecting IC functionality, there are other types of errors, likely to be caused by image artifacts: a component $C'_j \in \mathcal{C}_{M'}$ is a *noise* if $C'_j \cap C_i = \emptyset$ for every $C_u \in \mathcal{C}_{\mathbf{M}}$. Moreover, $C'_j \in \mathcal{C}_{M'}$ *contains a hole* if it contains a component of $\mathbf{M}'^{-1}(0)$ within its interior, while none of $C_i \in \mathcal{C}_{\mathbf{M}}$ does so.

Intuitively, noise corresponds to small additional artifacts in the IC images, e.g., caused by dust. Holes in the segmentation can result from lighting conditions, where parts of a metal line appear dark during image capture

In summary, we consider 4 main *error types*:

- open circuits
- short circuits
- noise
- holes.

For the ground-truth mask $\mathbf{M}$, a graph $G_i \in \mathcal{G}_{\mathbf{M}}$ usually has a tree structure, as it is the homotopy retract of a metal-line component, which by design, is unlikely to have holes inside. If this is true for each $G_i \in \mathcal{G}_{\mathbf{M}}$, we call $\mathcal{G}_{\mathbf{M}}$ *consists of trees*. Such a property is topological. In general, the 1*st Betti number* (Hatcher, 2001) is sufficient to encode such a property for a graph. For the simplified definition in the case of graphs, the 1st Betti number $b_1(G)$ counts the number of (closed) loops in a graph $G$. For example, $b_1(G) = 0$ if $G$ is a tree.

## A.2 ANALYSIS OF SEGMENTATION ERRORS

In this subsection, we discuss a result that gives conditions on segmentation errors, using the above theoretical framework. We keep the *notations and assumptions* of the previous subsection. For an error, we assume that it belongs to one of the types described in the previous section, which most commonly occurs in IC image segmentation.

**Theorem 1.** *If any of the below holds, then there is an error in the segmentation mask* $\mathbf{M}$:

(a) *The number of components* $m \neq m'$.

(b) $\mathcal{G}_{\mathbf{M}}$ *consists of trees and for some* $C_i \cap C'_j \neq \emptyset$, $b_1(G_i) \neq b_1(G'_j)$.

(c) *Let $K$ be an upper bound on the number of components for any symmetric differences, and assume the largest diameter of any component is bounded by $\epsilon' \leq \epsilon$. For some $C_i \cap C'_j \neq \emptyset$, there does not exist any construction of $G'_j$ such that $d_{GH}(G_i, G'_j) \leq K\epsilon'/2$.*

Notice that the three conditions are arranged in order of increasing difficulty of verification. (a) can be interpreted as matching the 0th Betti number of the segmentation mask, while (b) is the matching of the 1st Betti number of each graph component. Both of them are topological in nature, which are coarser. On the other hand, (c) matches the graph components metrically, which is a more refined condition.

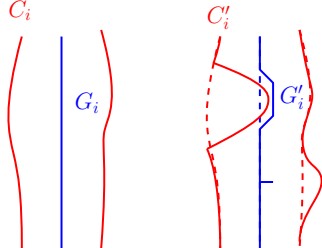

Figure 8: Due to possible small perturbations of $C'_i$, we modify $G_i$ accordingly depending on the perturbation type.

*Proof.* Suppose $m \neq m'$ and there is no noise, then for each $C'_j$, there is a $C_i$ such that $C_i \cap C'_j \neq \emptyset$. As $m \neq m'$, we have either of the following cases:

- For $j_1 \neq j_2$, $C_i \cap C'_{j_1} \neq \emptyset$ and $C_i \cap C'_{j_2} \neq \emptyset$.

- For $i_1 \neq i_2$, $C_{i_1} \cap C'_j \neq \emptyset$ and $C_{i_2} \cap C'_j \neq \emptyset$.

The former corresponds to an open circuit, as the diameter of some components of $C_i \backslash C_{j_1}$ is $> \epsilon$ by the $\epsilon$-separability of $\mathbf{M}'$. Similarly, for the latter case, there is a closed circuit, as the diameter of some components of $C'_J \backslash C_{i_1}$ is $> \epsilon$ by the $\epsilon$-separability of $\mathbf{M}'$. Hence, there is always an error if $m \neq m'$. This handles (a).

For the rest of the proof, we assume that $m = m'$. Consider condition (b). Re-order the indices if necessary, we assume that $C_i \cap C'_i \neq \emptyset$ for any $i \leq m$ and for some $i \leq m$, $b_1(G_i) \neq b_1(G'_i)$. As $G_i$ and $G'_i$ are the respective homotopy retracts of $C_i$ and $C'_i$, the 1st Betti numbers of $b_1(G_i) \neq b_1(G'_i)$ by the assumption, so are $b_1(C_i)$ and $b_1(C'_i)$ (Hatcher, 2001). Therefore, as $G_i$ is a tree and thus the $b_1(C_i) = 0$, we have $b_1(C'_i) > 0$. Hence, there is at least an error type of a hole in $C'_i$.

For (c), suppose the condition holds for an index $i \leq m$. We assume that there is no noise or holes in the segmentation mask $\mathbf{M}'$. We claim that there is an open or short circuit. The strategy is to modify $G_i$ to construct an $\epsilon$-approximation $G'_i$ of $C'_i$ such that $d_{GH}(G_i, G'_i) \leq \epsilon$. This will contradict the assumption.

For this, we need an equivalent definition of the Gromov-Hausdorff distance (Tuzhilin, 2020). For two metric spaces, a correspondence is a relation $R \subset X \times Y$ such that the projections (from $R$ to $X$ or $Y$) $p_X, p_Y$ satisfy: $p_X(R) = X$ and $p_Y(R) = Y$. The distortion of the correspondence $R$ is defined as:

$$\delta(R) = \sup\{|d(x, x') - d(y, y')| \mid (x, y), (x', y') \in R|\}. \tag{7}$$

Then, $d_{GH}(X, Y) = \inf_R \delta(R)/2$, where the infimum is taken over all correspondences $R$. In particular, if for some $R$, $\delta(R) \leq 2\epsilon$, then $d_{GH}(X, Y) \leq \epsilon$.

If there is no open or short circuit, then the symmetric difference $C_i \Delta C'_i$ has at most $K$ components, each with diameter bounded by $\epsilon'$. For each such component, we construct an $\epsilon$-approximation $G'_i$ of $C'_i$ by modifying a small segment $\ell$ of $G_i$ by one of the following two ways (see Fig. 8 for an illustration):

- Replace $\ell$ by a union $\ell'$ of small segments whose total length is within $\epsilon'$ of $\ell$, and set the linear correspondence between $\ell$ and $\ell'$.

- Add a small linear segment $\ell'$ of length at most $\epsilon'$ to be point $u_0$ on $G_i$. Let every point $\ell'$ correspond to $v$.

Now, for any pair of corresponding points $(u, v)$ and $(u, v')$ on $G_i \times G_i'$, the geodesic path (i.e., shortest path) connecting $u, u'$ differs from that of $v, v'$ by crossing at most $K$ modifications. Therefore, $|d(u, u') - d(v, v')| \leq K\epsilon'$. By (7) and its consequence, we have that $d_{GH}(G_i, G_i') \leq K\epsilon'/2$, which contradicts the assumption. $\square$

The *practical implication* of the result is that errors can be detected if we identify closely located components in $\mathbf{M}$ and $\mathbf{M}'$, while their corresponding graphs (i.e., approximations) are either topologically or metrically dissimilar. This prompts the use of GNN for error detection.

# B  LLM USAGE

We acknowledge the use of large language models (LLMs) as a general-purpose assistive tool in preparing this manuscript. Specifically, LLMs were employed to aid in polishing the writing, including refining grammar, improving clarity, and enhancing fluency of expression. LLMs were **NOT** used for generating research ideas, conducting analysis, or producing results. All conceptual contributions, theoretical developments, experimental designs, and interpretations presented in this work are entirely the responsibility of the authors.

