# OpenReview forum: "Explainable Error Detection in Integrated Circuits Image Segmentation via Graph Neural Networks"
_ICLR.cc/2026/Conference — ICLR 2026 Conference Withdrawn Submission_

### Official Review · Reviewer_zf7W · 2025-10-25

**Soundness:** 2
**Presentation:** 3
**Contribution:** 2
**Rating:** 2
**Confidence:** 4

**Summary:**

The paper presents an explainable error detection algorithm using a segmentation mask of integrated circuits (ICs). For that, each connected component of a segmentation mask is converted into a feature-annotated graph: the following features of component graphs
are observed to be useful: (i) locations (for positional proximity) and (ii) lengths of edges and turning angles (for similarity in topology and geometric shape). The experiments are performed using multiple datasets, and comparisons have been made with benchmark algorithms.

**Strengths:**

- The paper is easy to read and follow.
- The identification of faults in current complex integrated circuits is critical and essential.
- The proposed idea of utilising a graph to represent the connected component is trivial.

**Weaknesses:**

- While the paper and abstract suggest that the paper aims to develop an explainable error detection approach, the paper rarely focuses on interpretability. The section titled explainability talked about an ablation study instead of interpretability.
- The algorithms used for comparison are old and not state-of-the-art. The recent algorithm used to report values in Table 2 is from the year 2021.
- While the existing holistic approaches might be utilising full images, they might not be sensitive to the generated segmentation mask. Since the segmentation mask is one of the crucial components of the proposed approach, the analysis concerning faulty segmentation masks must also be explored.
- Is there any rationale behind using the parameters mentioned in the paper, such as deciding the corner pixel based on $90^o \pm \triangle^o$ and the number of incident edges? The motivation and theoretical justification are weak.
- Reporting the values in terms of F-1 is not sufficient. Moreover, are these F-1 values, weighted F-1 or normal F-1 (where equal weight is assigned to both classes)?
- Further, the generalizability of the proposed approach is significantly lower than the holistic approaches. Interestingly, when trained on a small dataset (A1 or A2), the ED-ResNet yields poor performance on S1 (or S2). There is no reason to justify this phenomenon that has been provided.

**Questions:**

- The explainability and interpretability of the proposed approach are weak.
- Recent algorithms must be used for comparison.
- The analysis concerning faulty segmentation masks must also be provided.
- The motivation and theoretical justification of the parameters used in the proposed approach are required.
- The results must be reported using either weighted F-1, and precision/recall/AUROC must also be reported.
- Generalizability of the proposed approach is also a major concern.

---

### Official Review · Reviewer_4ny9 · 2025-11-01

**Soundness:** 2
**Presentation:** 3
**Contribution:** 2
**Rating:** 4
**Confidence:** 5

**Summary:**

The paper addresses automated detection of segmentation errors in IC SEM masks by converting each connected component of a binary mask into a feature‑annotated graph and classifying components with a custom Edge‑Aware Graph Isomorphism Network (EA‑GIN). Node features include normalized coordinates, node type (end/junction/corner), and incident angles. Edge features summarize geometry and mask width statistics along skeletonized paths. Classification is performed binary (normal/abnormal). Experiments on four proprietary datasets (A1, A2, S1, S2) report strong F1 on in‑distribution and cross‑dataset tests versus several GNN baselines (GIN, GINE, EGAT, CensNet) and a CNN image‑level baseline (ED‑ResNet). The paper also argues for 'explainability' via component‑level localization and provides a topology/metric framework (Appendix) motivating outlier detection of graph components.

**Strengths:**

a. Clear problem formulation tailored to IC metrology

The paper motivates that IC foreground structures are defined more by topology than texture and thus benefit from a graph abstraction rather than holistic CNNs. It formalizes the goal as identifying 'unseen' component graphs relative to ground‑truth distributions and provides a pipeline from mask to skeleton to graph with explicit node and edge features (Figs. 3–5, Sec. 3.2)

b. Simple architecture with edge features

EA‑GIN augments GIN by turning edge features into attention‑like weights for message passing (Eqs. 3–5), a pragmatic design aligned with the task’s geometry (lengths, widths, turning angles). The model is compact (3 layers, small MLPs) and achieves large gains over plain GIN in all datasets (Table 2), highlighting the value of edge features.

c. Convincing performance on several datasets

On image‑level error detection, EA‑GIN reaches F1=0.99 on A1/A2 and 0.90–0.92 on the more complex S2/S1 datasets, outperforming or matching GNN baselines and often surpassing the CNN baseline under cross‑dataset generalization (Table 2–3). The cross‑dataset results (e.g., train on A1/A2, test on S1/S2) are particularly interesting given limited graph diversity in ICs (Table 3).

d. Explainability via component‑level flags

Component graphs with the top‑κ abnormal scores can localize errors (κ≤3 yields F1 close to 1.0 for A1/A2 and ≥0.97 for S2, Table 4, Fig. 7). This is useful for manual reviews and aligns with the thesis that errors are local.

**Weaknesses:**

a. Uncertainty and Robustness

The authors report F1 scores without providing variance estimates, confidence intervals, or clarifying whether the metric is computed as micro-, macro-, or weighted F1. This is highly problematic given the extreme class imbalance (e.g., A1: 190 positives vs. 31k negatives) and makes the reported performance sensitive to split choice. Stratified, folded cross-validation or repeated runs with mean, std and confusion matrices have to be reported for statistical reliability.

b. Cross-Dataset Performance Anomalies

Results in Table 3 show anomalies: models trained on smaller datasets (A1, S1) sometimes outperform those trained on larger datasets when tested cross-domain (e.g., A1->A2 = 0.9962 vs. A2->A2 = 0.9953; S1->S2 = 0.9155 vs. S2->S2 = 0.8990). This suggests instabilities or overfitting, yet no diagnostic evidence (learning curves, calibration plots) is provided. It further highlights the need for uncertainty reporting.

c. Baseline Comparison

While as a CNN-based method it is a relatively weak baseline, ED-ResNet consistently outperforms EA-GIN on complex datasets (S1 and S2), achieving a near-perfect F1 (e.g., S2: 0.9951 vs. 0.8990). This contradicts the claim of broad superiority for the graph-based approach and indicates that EA-GIN may struggle with high-density layouts.

d. Explainability and Evaluation

While the method claims interpretability, evidence is qualitative (visual examples) without quantitative attribution or per-error-type breakdown.

e. Missing Architectural and Computational Details

Hyperparameters (optimizer, learning rate, batch size, epochs) are not reported. Runtime and memory benchmarks for graph conversion and EA-GIN inference are also missing.

f. Minor Typos and Terminology Issues

Inconsistent terminology such as 'metal line/metalline/meta-lines' and grammatical errors appear throughout (e.g. 'For the high-level idea [...]'). Figure captions could be more precise, and definitions should precede tables for clarity (e.g. Table 4).

**Questions:**

The approach is conceptually strong and shows promise on simpler datasets, but the lack of uncertainty quantification, anomalies in cross-dataset generalization, weak performance on complex cases, and incomplete evaluation undermine confidence in robustness and generality. Major revision recommended.

---

### Official Review · Reviewer_MHzX · 2025-11-02

**Soundness:** 3
**Presentation:** 2
**Contribution:** 3
**Rating:** 4
**Confidence:** 4

**Summary:**

The paper proposes an error detection framework for integrated circuit (IC) image segmentation using Graph Neural Networks (GNNs). The overall framework handles the problem by converting each connected component of a segmentation mask into a feature-annotated graph, enabling localized reasoning and error identification through graph classification. Their proposed method, called Edge-aware Graph Isomorphism Network (EA-GIN), incorporates both node and edge features to enhance detection accuracy and interpretability. Experiments across four IC datasets demonstrate that EA-GIN achieves robust, generalizable, and interpretable error detection at the component level, outperforming several benchmark models.

**Strengths:**

• Originality - The work aims at solving a critical problem in IC diagnosis. The way of transforming the task into GNN classification is innovative.
• Explainability: The graph-based approach allows for precise localization of segmentation errors, making the model’s decisions more interpretable for practitioners.
• Component-level Detection: By focusing on individual connected components, the method can pinpoint specific error regions, which is valuable for quality control in IC manufacturing.
• Robustness and Generalization: EA-GIN demonstrates strong performance across diverse datasets and imaging conditions, and it generalizes well even when trained on smaller datasets.

**Weaknesses:**

• Clarity can be improved: The way of explaining mathematical/topological terms can be improved for better readability.
• Complexity of Graph Construction: The paper lacks the details of converting images into binary masks. The process of converting segmentation masks to graphs and extracting meaningful features may introduce additional computational overhead and implementation complexity.
• Dependence on Quality of Segmentation Masks: The effectiveness of the approach relies on the quality of the initial segmentation; significant errors or artifacts in the mask could affect graph construction and subsequent error detection.
• Limited to Structural Errors: While the method excels at detecting topological anomalies (e.g., open/short circuits), it may be less effective for errors that do not manifest as structural changes.
Potential Overfitting in Large Datasets: The paper notes that training on larger datasets can sometimes lead to overfitting, suggesting that further work is needed to improve model regularization and scalability. The paper lacks the detailed discussion of how to prevent overfitting to the dataset.

**Questions:**

What is the method used to convert images into binary masks?
Instead of relying on CV to enhance the overall performance, why didn't you enhance the model and prevent overfitting issues?
What are the sources of the database used in the experiments? What technology node is associated with them?

---

### Note · Authors · 2025-11-12

I have read and agree with the venue's withdrawal policy on behalf of myself and my co-authors.